# Self-Assembly at a Macroscale Using Aerodynamics

Yi Liu [1,2], Yuting Chen [2,3,4], Xiaowu Jiang [1], Qianying Ni [1], Chen Liu [2], Fangfang Shang [2], Qingchao Xia [1,2] and Sheng Zhang [1,2,*]

1 School of Mechanical and Energy Engineering, Ningbo Tech University, Ningbo 315100, China; liuyilulu@nit.net.cn (Y.L.); jiangxiaowu115@gmail.com (X.J.); ni_i@foxmail.com (Q.N.)
2 Ningbo Innovation Center, State Key Laboratory of Fluid Power and Mechatronic Systems, School of Mechanical Engineering, Zhejiang University, Ningbo 315100, China; chenyuting@stumail.ysu.edu.cn (Y.C.); liuchen@nit.zju.edu.cn (C.L.); shangfang@nit.zju.edu.cn (F.S.); mynameisxia@zju.edu.cn (Q.X.)
3 School of Mechanical Engineering, Yanshan University, Qinhuangdao 066004, China
4 Hebei Heavy Machinery Fluid Power Transmission and Control Lab, Yanshan University, Qinhuangdao 066004, China
* Correspondence: szhang1984@zju.edu.cn

**Abstract:** Intuitive self-assembly devices are of great significance to the emerging applications of self-assembly theory. In this paper, a novel intuitive device with an aerodynamic system is fabricated for the self-assembly experiment. Table tennis balls were used as the objects to be assembled during the self-assembly process. To understand more about the system, two experiments were designed—the directed assembly experiment was conducted to organize a specific structure and to explore the influences of environmental variables, and the indirect assembly experiment repeated with the "bottom-up" self-organization process and expressed the characteristics of "the optimization" and "the emergence" in the self-organization process. This article expressed a novel self-assembly approach at a macroscale and created a new choice or idea for the structural design and the optimization method.

**Keywords:** self-assembly; directed self-assembly; indirect assembly; self-assemble device

## 1. Introduction

Self-assembly (SA) is spontaneous organization that arranges tiny components into desired structures without human intervention [1]. Originating from the spontaneous combination of selected molecules or particles into structures [2,3], self-assembly technology is akin to shaking a box of tiny components similar to Lego bricks. With specific shapes or functions, structured materials carry away unsuitable substances and arrange others that are desirable. It is a cost-effective "bottom−top" process [4], where lives are formed through the self-assembly process, and solutions to many research fields are expanded [5]. However, programming the individual pieces into the desired structure remains a major challenge. Forces, e.g., capillary action force [6], electrokinetic flow force [7], hydrophilic (hydrophobic) action force [8], magnetic force [9], and acoustic drive force [10] have been utilized for self-assembly driving. Macromolecular building in self-assemble research includes ellipsoids [11], monolayers [12], patchy particles [13], polyhedral [14], latex particles, and various particle shapes built from nanoparticles [15]. At the same time, different physical environments, such as magnetic fields, electric fields [16], turbulent air, or agitated fluid, have served as the media in these scenarios recent years.

Ulukan et al. proposed that self-assembly will be an important element of manufacturing in the next century [17], which painted a picture of materials experiencing spontaneous coalescence rather than top-down component construction processes. Striving for scalability, adaptability, and reconfigurability, self-assembly is where the conducted parts can build themselves [18–20]. Founded on the research and analysis of complex systems in other domains, e.g., biology and physical chemistry, the phenomenon of self-organization of macro functional structures is possible. This process is realized by the local interaction of low-level

particles. For example, in biological systems, self-assembly construction exists in cellular division as the body's ability to grow and repair. Various elements in the system compete and cooperate with a high efficiency, higher-order structures are built, the effect of various resources is maximized, and their functionality emerges autonomously [21]. The essence of the mechanical assembly structure is also a high-level complex functional structure formed through the interaction of local energy, materials, and information. Well-designed assembly product systems are normally nonlinear and complex with a certain organization. With the improvement of product complexity, the organization will be strengthened usually. As a result, the orderly evolution rule needs to be explored, and the "optimization" and "emergence" mechanism contained in self-assembly is the exact endogenous internal force that is needed to solve the collaborative order problem of complex systems.

The majority of related research has been conducted at a microscale in order to explore the molecule assemble method. However, at a larger scale, especially in the mechanical assembly field, Tibbits et al. conducted a limited amount of research [22], where molecular self-assembly through tangible physical models was demonstrated in 2007. The geometry and material components were based on various molecular structures, such as the tobacco plant virus, ferritin protein, and catechol dioxygenase enzyme [23]. In 2012, a container was designed to tumble the units inside, and it was discovered that units could be self-assembled into structures at a furniture scale [24]. In 2014, formations of crystal growth were imitated in a fluid environment. In the same year, research called the "self-assembly chair" was conducted in fluid environments to demonstrate the process of irregular asymmetric components. Other research, such as "aerial balloon assembly", investigated the possibility of large-scale self-assembly in aerial environments by using lightweight structures. Self-replicating spheres explored the processes of growth and division through the agitation of simple spherical units in 2015 [23]. It has been proposed that a self-assembly system must include four components: parts for assembly, links for connection, a container for accommodation, and the target for the final assembly goal [21]. Furthermore, the driving energy of the assembled system needs to be calculated: too much driving energy will cause the links to break; however, too little driving energy will not allow the parts to move together and be pinned to each other [25].

Although some researchers have already conducted self-assembly experiments at a microscale, a macroscale self-assembly method is still required. Compared with the research mentioned above, it is still of great significance to design a macroscale self-assembly device with the ability to control the environmental variables. Moreover, the purpose of previous research was to realize a self-assembly process of a specific structure; however, in this paper, the self-assembly device introduced is not limited to this—it also represents the "optimization" and "emergence" mechanism by experiments. In this research, the designed experimental device is more suitable for laboratory circumstances, and the whole device can easily fit onto a normal desk. The designed pneumatic self-assembly device controls the airflow force as the driving implement. The parts for assembly are divided into two groups: parts of the directed assembly experiment and parts of the indirect assembly experiment. Considering the weight and volume limitation caused by the airflow force and the size of the assembly device, table tennis balls were chosen as the assembly object, and the magnetic force served as the links. Both the direct self-assembly process and the indirect assembly process were performed so as to understand more about the device and the self-assembly process, and the results of these experiments are discussed.

## 2. Design of the Self-Assemble Device

The table tennis ball resembles the natural atomic nucleus, as it is light in weight and round in shape. In this research, the table tennis ball we selected was about 40 mm in diameter and 3 g in weight. Magnets were placed on the ball surfaces and served as a link pin, and the linkage was connected through the attraction of the magnetic force. The magnetic field intensity of the magnet surface was about 2000 G, and the magnetic field dropped to 0 G when it was 27 cm away from the magnet surface. Similar to the chemical

bonds within the molecules, the magnetic force was relatively strong, but it still had the possibility to be broken by the impact of external forces. Furthermore, the specific molds used to locate the magnets were accurately designed and fabricated using a 3D printer. Holes were designed to be in the center of each surface of the mold, and the magnets were placed and glued to the balls' surface. A set of molds with a tetrahedral structure is described in Figure 1A. When the balls are assembled, the linked structure is similar to the methane molecule (Figure 1B).

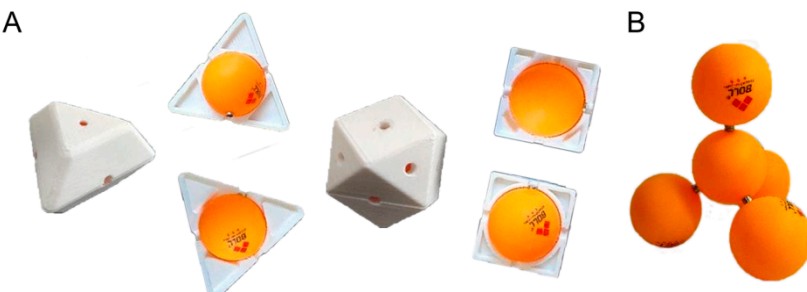

**Figure 1.** The molds and the structure of a tetrahedron. (**A**) The tetrahedral molds with locational holes for the magnets. (**B**) The ideal assembled tetrahedron structure is similar to the methane molecule.

Compared with liquid agitated sinks or vibrated containers, the device in this paper is more stable, and is more convenient for arrangement and observation using a multi-angle camera, as shown in Figure 2A (Raspberry Pi Camera V2.1, Raspberry Pi Foundation, Wales, England). The speed of the airflow can be adjusted easily and is fast to respond. The overall size of the designed device is about 65 cm × 25 cm × 50 cm, which contains three parts: an assembly cage in the upper left, a numerical-controlled airflow generator on the lower left, and the host computer and electrical cabinet on the right (Figure 2D). The assembly cage is a funnel-shaped closed space surrounded by nets with 8 mm diameter holes, which are 15 cm ×10 cm at the bottom and 24 cm × 24 cm at the top. The nets could limit the movement of the testing objects that need to be assembled, but not hinder the passage of the airflow too much in the meantime. The camera is placed directly above the assembly cage to monitor the assembly process in the cage (Figure 2D).

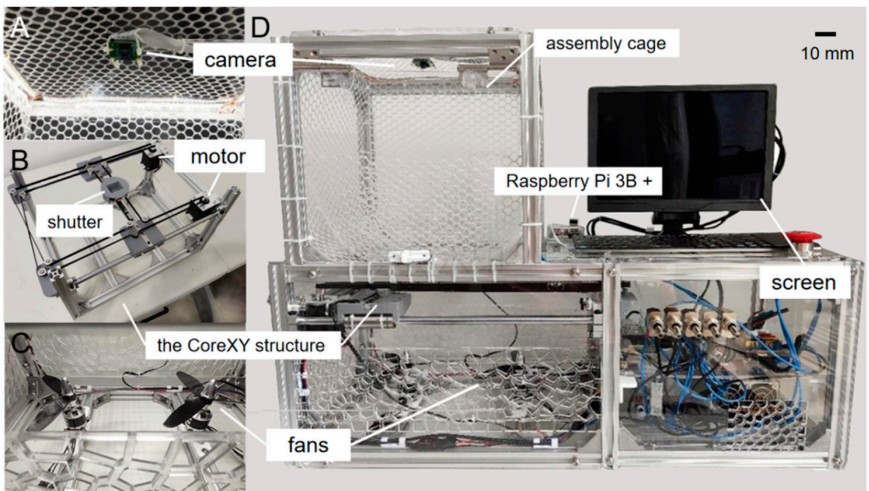

**Figure 2.** The overall self-assembly device and its components. (**A**) The Raspberry Pi Camera V2.1 is placed on the top for the multi-angle observation, and is facing towards the assembly process. (**B**) The CoreXY structure uses two motors to control the shutter's two-direction movement. (**C**) The two fans are placed in the lower part, each of which is composed of a model aircraft motor and a nylon three-blade propeller. (**D**) The overall assembly device.

The numerical-controlled airflow generation device, which is mainly composed of two fans and one numerical control shutter, could generate controllable and adjustable vertical airflow to be the power supply of the assembly process. In Figure 2B, the shutter moves in two directions, generating a low-pressure region above, and it is controlled by a "CoreXY structure". The "CoreXY structure" is widely used in commercial 3D printers, with two motors acting at the same time, which makes the shutter control more stable than for the single one. When the two motors rotate in the same direction, the shutter can move in one direction (this direction is defined as "X"). On the contrary, when the two motors rotate in the reverse direction, the shutter can move in another direction perpendicular to the "X" (this direction is defined as "Y"). Meanwhile, this structure has a sufficient amount of open-source firmware, such as GRAB firmware, which is used to program the position of the shutter in this paper. The shutter can be simply controlled by sending a series of Gcode after the configuration, as the Gcode instruction and protocol can be completely supported by the GRAB firmware. In Figure 2C, the two fans that are placed in the lower part can provide the airflow, with a maximum velocity of 15 m/s, and each one is composed of a model aircraft motor (Yinyan X2212 980 KV, YINYAN MODEL TECH. LTD., Shenzhen, China) and a three-blade propeller (GEMFAN 6045, GEMFANHOBBY Co., Ltd., Ningbo, China).

To ensure that all of the components (GRAB firmware, OpenCV, and model aircraft motors) can be unified, Raspberry Pi 3B+ is selected as the main controller in this paper. This is a micro-motherboard based on ARM and is only slightly larger than a credit card (Figure 2D). Raspberry Pi 3B+ has all the basic functions of the personal computer (PC). Other parts include an SD/MicroSD card as the hard drive; a 1/2/4 USB interface; a 10/100 Ethernet interface; a video analog signal television output interface; and an HDMI video output interface for the connection of the keyboard, a mouse, a cable, and a monitor in Raspberry Pi 3B+.

## 3. Experiments

### 3.1. The Directed Assembly Experiment

A mechanical assembly process will be meaningless if the desired results are not obtained. Therefore, an assembly experiment based on the self-assembly method was designed in this paper to assemble a specifically designed tetrahedral structure by specifically designing each connecting link, and the environmental factors could be changed and controlled. The structure of the tetrahedral was composed of one central ball and four sub-balls. The central ball possessed four connecting sites with the "N" magnet polar on the outside, while there was only one connecting site with the "S" magnet polar on the outside of each sub-ball. The balls with the same characteristic pole on the outside could not be connected. As a result, this structure ensured a specific match between the central ball and the sub-balls.

This experiment not only repeated a tetrahedral structure self-assembly process and demonstrated its feasibility, but it also studied the influences of environmental variables on the assembly process and the results. The environmental variables included the number of sub-balls, whether to change the wind speed, and whether to move the shutter during the assembly process. Three groups of experiments were designed to study the effects of the above environmental factors on the final assembly time. The number of sub-balls in the three experimental groups varied from four to six. The change in wind speed was realized by outputting a strong wind and a weak wind in turn within a pulse period (5 s). In the control experiment, the fans output a constant wind speed of 8 m/s. Moreover, in the experiment with a moving shutter, the shutter made a constantly reciprocating movement in the lower part of the assembly cage for a period of 5 s. The camera was used to record the assembly process with a frame rate of 20 fps (frames per second).

### 3.2. The Disordered Assembly Experiment

Most atoms are more stable only they form molecules, which is a result of natural evolution. In chemistry, chemical bonding is a pure quantum effect that is caused by the overlap of electronic densities, which determines the stability and structure of molecules. However, in some general courses of chemistry, this is simplified by judging whether the external electron number of an atom is appropriate or not. For example, a hydrogen atom only has one external electron, so it needs to share one electron with another atom to achieve a stable structure with two electrons around it. An oxygen atom has six external electrons around it, so it needs to gain two shared electrons from other atoms to achieve a stable structure with eight peripheral electrons. In this paper, a table tennis ball with one magnet site on its surface is similar to a hydrogen atom, as its external magnet pole also has a strong desire for the magnetic poles on the surface of other atoms. As a result, it is no longer "stable" by itself. Similarly, a table tennis ball with four magnet sites on its surface is similar to a carbon atom, with four "electrons" that are eager to be shared. The number of magnet sites on the surface of a table tennis ball mimics the covalent bond in chemistry.

To simulate the assembly process of a mixed natural environment with many kinds of atoms, this experiment aimed to explore the assembly law of balls further. Eleven balls with randomly arranged magnets on the surface were placed into the assembly cage. Among these balls, one ball had four magnets on the surface, two balls had three magnets on the surface, two balls had two magnets on the surface, and six balls had one magnet on the surface, which means that the final results will be unknown and stochastic. Each assembly experiment lasted for 3 min and was repeated three times in the environment with a changing wind speed and a moving shutter.

## 4. Results and Discussions

### 4.1. Results and Discussions of the Directed Assembly Experiment

Assembling a specifically designed tetrahedral structure (through the specific design of each connecting link), the directed assembly experiment is to test the influences of the environmental factors towards the assembly time. Then the specific assembly time of those 3 groups of experiments was tabulated in Table 1:

**Table 1.** The specific assembly time for the three groups of experiments.

| Environmental Variables / Number of Sub-Balls | Changing Airspeed with a Moving Shutter (s) | Constant Airspeed with a Moving Shutter (s) | Changing Airspeed with a Stationary Shutter (s) |
|---|---|---|---|
| 4 | 72 | 82 | 98 |
| 5 | 59 | >150 | 125 |
| 6 | 53 | >150 | 91 |

As shown in Table 1, with a changing wind speed and a moving shutter environment, as the number of sub-balls increased, the assembly process was completed more quickly. This result is similar to the concept of the reactant concentration in a chemical reaction process, that is, the higher the concentration, the faster the chemical reaction. The constant wind speed prolonged the assembly process, especially in the circumstances of five or six sub-balls. Regardless of the number of sub-balls and whether the shutter moved or not, only in the circumstances with a changing wind speed could the assembly process be finally completed, which demonstrates that a stirring process could accelerate the reaction process. Figure 3 shows the self-assembly process in the circumstance of four sub-balls with a changing wind speed and moving shutter.

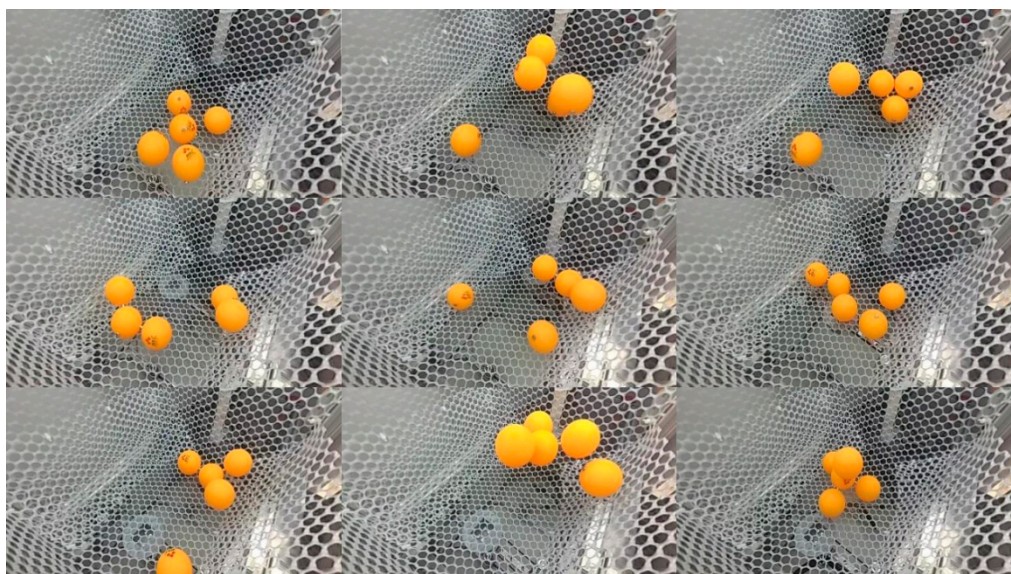

**Figure 3.** The assembly process of four sub-balls during a changing airspeed.

For a further analysis, this paper used the feedback from the observational camera and found that even after specifically designing the tetrahedral connection, the wrong connection would still exist (refer to the red wireframe in Figure 4). Sub-balls with the same polar magnet on the outside could be attracted and connected on the side surface of the magnet cylinder, where the magnetic field intensity was about 800 G. These incorrect connections did not have much chance of being corrected in a constant wind speed environment. However, under the circumstances of changing wind speed (especially in the strong wind stage), they could be corrected soon after being produced (refer to the blue wireframe in Figure 4), and the assembly process would be completed, which means that the assembly errors could be corrected by the change in the environment. In the real environment, atoms collide with each other randomly and indirectly. A temporary unstable connection may be formed as well, but usually this will not be prolonged. Only after screening the environmental parameters will the results be stable and sustainable.

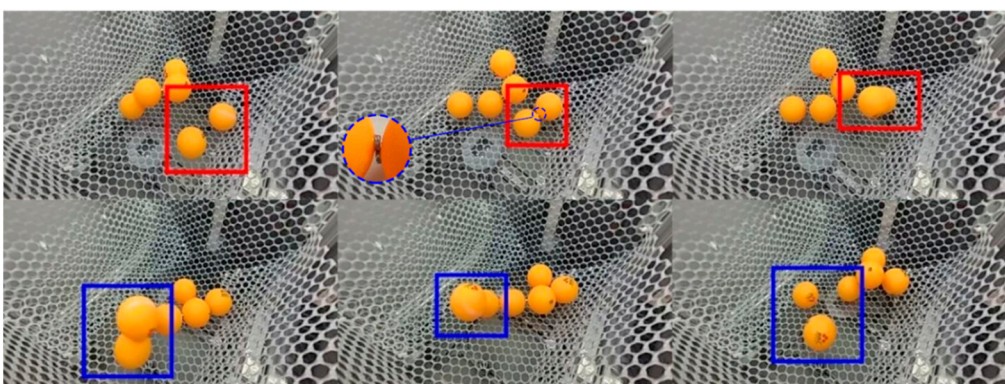

**Figure 4.** The error connection's formation and correction. The red wireframe is the forming process, and the blue wireframe is the fracture process.

This experiment proves that increasing the number of sub-balls can speed up the assembly process under suitable circumstances, which is similar to the concept of the reactant concentration in a chemical reaction process. Incorrect connections could appear in the assembly process, but they could be corrected by modifying environmental variables such as the airspeed and direction. Thus, this experiment not only repeated the assembly process of a tetrahedral structure, but also validated the principle that the stirring process

and an appropriate number of spare parts could accelerate the reaction process, which provides ideas for the optimization direction of the assembly process.

### 4.2. Results and Discussions of the Indirect Assembly Experiment

To simulate the assembly process of a mixed natural environment with many different kinds of atoms, the experiments consisted of 11 balls with randomly arranged magnets on the surface, which lasted for 3 min and were repeated three times in the environment with a changing wind speed and a moving shutter. The following results (those composed of several balls) of the relatively stable connection were discovered (Figure 5). Although the generated structure results varied each time, most of these constructed structures could withstand the changing environment well. Among these results, there were some essential differences between the chemical bond and the magnet skeletal bond, and between the shape of the molecules and the connected balls (the form of the electron density cloud was different from the magnet sites). If we only paid attention to the table tennis balls, which mimicked the atomic nucleus and contributed to the final structure, the structures generated from these experiments were similar to some common molecules, e.g., the carbon dioxide ($CO_2$)/water ($H_2O$) structure and the hydrogen ($H_2$)/oxygen ($O_2$) structure in Figure 5A, the ammonia structure ($NH_3$) in Figure 5B, the hydroxylamine ($NH_2OH$) structure in Figure 5C, the ethylene structure ($CH_2CH_2$) in Figure 5D, the methane (CH4) structure in Figure 5E, and the methanol ($CH_3OH$) structure in Figure 5F. By observing these stable assembly results, most of the stable structures could find prototypes in reality and most of those prototypes would be gas or volatile liquids. These stable structure results were the outcomes of specific experimental conditions and environments. Furthermore, the assemblies were similar in structure to each other. Table tennis balls with three or four magnet poles were more likely to be the center of a structure because of their greater number of poles, which are similar to the positions for the carbon atoms, nitrogen atoms, and oxygen atoms in many molecules. This experiment demonstrated that the molecular structures that exist on Earth must be highly adaptable to a disturbed airflow environment.

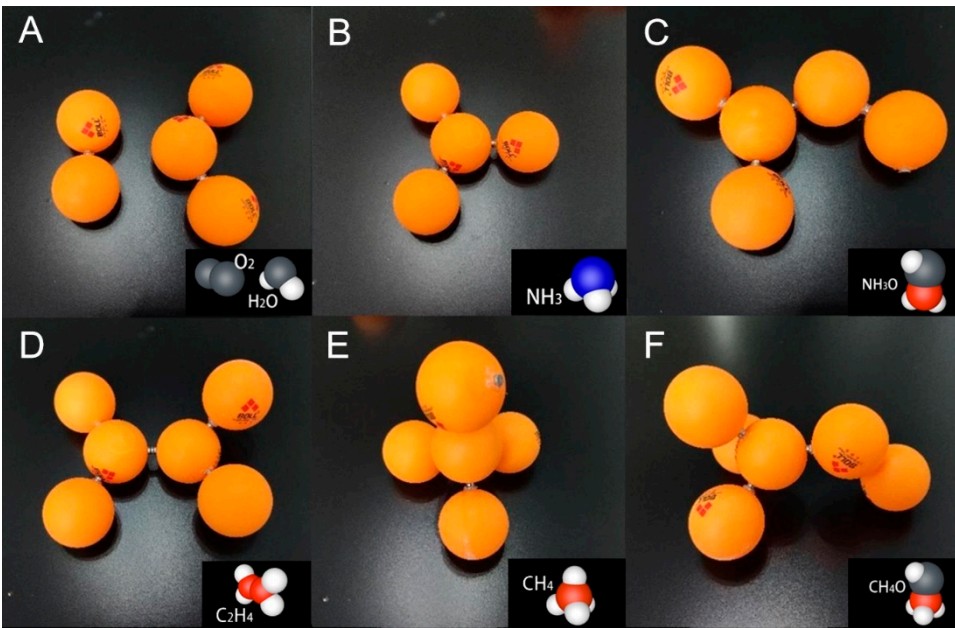

**Figure 5.** The stable connected structure results. (**A**) The water ($H_2O$)-like structure result and the hydrogen ($H_2$)/oxygen ($O_2$)-like structure result. (**B**) Ammonia ($NH_3$)-like structure result. (**C**) The hydroxylamine ($NH_2OH$)-like structure result. (**D**) Ethylene ($CH_2CH_2$)-like structure result. (**E**) Methane ($CH_4$)-like structure result. (**F**) Methanol ($CH_3OH$)-like structure result.

In this experiment, it was not likely that all of the balls would be connected to a single structure. As can be seen in Figure 6, a complex unstable structure with all of the balls

assembled was formed at one point (the lower left quarter of Figure 6), but it soon broke. In the atmosphere of the Earth, although pollen-like complex molecules exist, the gas on the Earth is still mainly composed of 78% nitrogen, 21% oxygen, 0.94% rare gas, 0.03% carbon dioxide, and 0.03% other gases. Bio-macro-molecules (such as protein, nucleic, lipid, and sugar) are more adaptable to a relatively stable, high-viscosity, and low-speed environment.

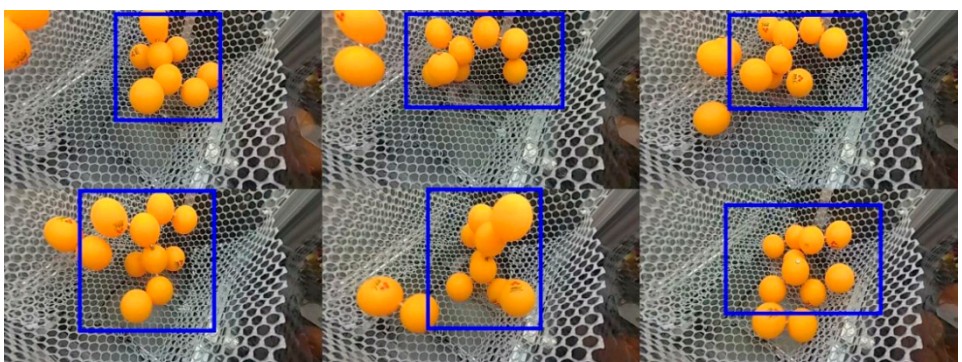

**Figure 6.** The breaking process of an unstable connected structure (which is highlighted by the blue wireframe).

The phenomenon of optimization was presented in this assembly experiment, that is, the stable structures were preserved, and the unstable structures were eliminated. The diversity of the final assembly results were generated by the connections of the table tennis balls without a specific design, which means that this whole experimental facility could be used to find stable structures that satisfied the environmental requirements.

### 4.3. Other Results and Discussions

Moreover, this paper applied OpenCV for the color feature processing and to obtain the color density thermodynamic map. The original image (Figure 7A) was primarily converted from the RGB color space to the HSV color space for the extracted and binarized process (Figure 7B). Then, the extracted binarization mask was attenuated and superimposed on the time axis, and the superimposed image was mapped into ".dst". As a result, the color density thermodynamic map was obtained (Figure 7C). According to the color density thermodynamic map, it can be seen that the probability of the spatial distribution of the table tennis balls in the assembly space was different.

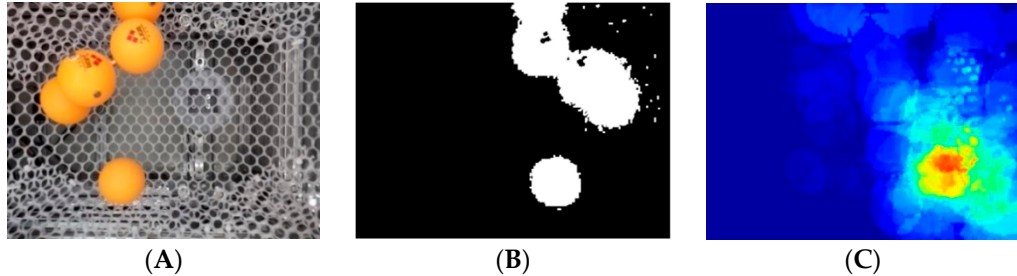

| (A) | (B) | (C) |

**Figure 7.** (**A**) The original image. (**B**) The color extracted image. (**C**) The color density thermodynamic map.

The fans blowing the ping-pong balls up can be simplified by the problem of aerodynamics. We believed that the airspeed would decrease with the increase in distance from the outlet of the air. In the beginning, the ping-pong balls were placed freely at the bottom of the assembly cage and the velocity of the balls ($v_s$) was 0 m/s. As the fans turned on and the wind blew up from the lower part, each ball experienced a wind pressure force ($F_p$). $F_p$ was produced by the wind pressure and was directly applied to the upwind side of the ball (Formula (1)), which is the classical application of Bernoulli's principle. Moreover, a low

fluid velocity produced a pair of separated and steady vortices on the downwind side of the ball, and the effect of the vortices should also be considered when counting the ball's drag force (Formula (2)). Formula (3) is the calculation equation of the Reynolds number.

$$F_p = \omega_0 Ag = 0.5 \times rho \times v_f^2 \times A \tag{1}$$

$$D = Cd \times 0.5 \times rho \times v_f{}^2 \times A \tag{2}$$

$$Re = \frac{|v_f - v_s| \times d_s \times rho}{\mu_f} \tag{3}$$

$C_d$ is the drag coefficient, $D$ is the drag force considering the vortices' effect, *rho* is the air density, $v_f$ is the fluid velocity, $d_s$ is the ball's diameter, and $\mu_f$ is the fluid kinetic viscosity. Both the wind pressure force ($F_p$) and the drag force ($D$) were positively correlated with the fluid speed. The air kinetic viscosity is $\mu_f = 17.9 \times 10^{-6}$ Pa·s, $d_s = 0.04$ mm, and the air density is *rho* = 1.18 kg/m$^3$ at 20 °C, 101.325 kPa.

When the velocity of the balls ($v_s$) was 0 m/s, and the velocity of the wind ($v_f$) ranged from 1 m/s to 15 m/s after turning on the fans, according to Formula (3), the Reynolds number ranged from 3000 to 40,000. Refer to the $C_d$ -*Re* curve graph from the website of NASA or other relative articles [26,27]; as the surface of table tennis balls is smooth, $C_d \approx 0.5$. Moreover, the ping-pong ball still experienced the effect of gravity, which was about 0.03 N, always in a straight-down direction. According to Formula (1), if $F_p \geq 0.03$ N, then $v_f \geq 6.4$ m/s. According to Formula (2), if $D \geq 0.03$ N, then $v_f \geq 9.0$ m/s. In the directed assembly experiment with a constant airspeed at $v_f = 8$ m/s ($v_s = 0$ m/s), according to Formula (3), the Reynolds number = 21,000, and thus $C_d = 0.5$. Then, according to Formula (1), the lift force was slightly greater than the gravity of the table tennis ball (0.03 N), but according to Formula (2), the lift force was smaller than the gravity. In the directed assembly experiment at an airspeed of 8 m/s, the assembly force was appropriate for the assembly process, which was not strong enough to break all the links, but also was too weak to connect to each other. This phenomenon contradicted the formula, but was consistent with the Bernoulli equation, which could be caused by the uneven distribution of the airspeed, the increasing function of the shutter, and the net towards the airflow velocity, or by other forces that have not been considered.

Where there is spinning, the Magnus Effect occurs. Because of the uneven airspeed distribution in the assembly cage, the airspeed on the two sides of the table tennis ball could be different, which will result in the ball spinning. The rotation of the ball means that one-half of the ball is in the same direction as the airflow, while another half of the ball is moving in the contrary direction. The same direction of movement will produce a region with low pressure, while the other side will create a region with high pressure, which will create a centripetal force pointing to the low-pressure region. The ball will deviate from its original direction and move in a circular arc. Formula (4) is the calculation equation for the centripetal force. Formula (5) is the calculation equation for the radius of the circular arc. Formula (6) is a common equation for static friction. Parameter $C_L$ is the lift coefficient, $b$ is the radius of the ball, $s$ is the angular speed of the ball, *rho* is the density of the air, $v$ is the relative velocity of the ball compared with the air, $R$ is the radius of the curve, $m$ is the mass, $\mu$ is the friction coefficient, $f$ is the friction force of a statistic subject, and $\pi$ is the standard value of 3.1415. In the directed assembly experiment with a constant airspeed at $v = 8$ m/s, it was assumed that the angular speed $s = 2\pi$ rad/s, $C_L = 0.15$ [27], $\mu \leq 1$, and according to Formulas (4) and (6), $L = 0.0037$ N, which was greater than $f$. Therefore, if the ball rotated under a vertical upward airspeed of 8 m/s, the Magnus force would make the ball move horizontally. Meanwhile, if the rotating ball was blown into the air, the radius of its motion track can be referred to using Formula (5).

$$L = \frac{4}{3} \times C_L \times (4 \times \pi^2 \times b^3 \times s \times rho \times v) \tag{4}$$

$$R = \frac{3 \times m \times v}{16 \times C_L \times rho \times s \times b^3 \times \pi^2} \tag{5}$$

$$f = \mu \times m \times g \tag{6}$$

However, the purpose of this research was to compile a qualitative study of the self-assemble law. As a result, the limited weight and the volume of the assembly objects caused by the driven wind force and the uniform flow-field quality were not vital. Moreover, the connection method was limited to using magnet forces on a centimeter-scale, which also limited the specific connection design possibilities (the magnet had only two poles and the object was small). Therefore, these existing constraints caused a lot of limits towards the assembly objects of this device, so it seems that there will be few practical applications for this assembly method. In addition, the incorrect connection mentioned in the first experiment of this research was not a significant operational error for the directed assembly process, as it can be eliminated by embedding the magnet into the balls or reducing the thickness of the magnet. Moreover, with the development of material science, the emergence of some dynamic programmable materials will bring about more possibilities for its utility application. The scale of this device can also be designed to be larger in order to optimize the specific connecting method, and to better meet the larger-scale application needs. Moreover, concerning the purpose of conducting a directed assembly experiment, the incorrect connection was meaningful and informative as it simulated a temporary unstable connecting formation in the real environment as the atoms collided with each other randomly and non-directly.

## 5. Conclusions

In this research, a new platform was designed for the self-assembly experiment and table tennis balls were used as the assembly parts. This new platform enabled the experiments to adjust the environment variables of the self-assembly process. The directed assembly experiment with specific connection designs aimed to promote the final specific result and to search the factors influencing the assembly time through a series of adjustments of environmental variables, which is a "top-down" process. The indirect assembly experiments with unspecific connection designs, which is a "bottom-up" process, aimed to obtain adaptable results for the turbulent pneumatic environmental conditions, which simulated and visualized the real random assembly process with a new human-scale device.

Through these two experiments, the following rules are summarized: (a) The assembly process can be sped up by increasing the number of spare assembly parts in a suitable environment, which repeats the concept of the reactant concentration from a chemical reaction process. (b) The incorrect connection can be corrected through suitable regulation of the environment, which is similar to the temporary unstable incorrect connection formed by the random collision of the atoms, which is also broken by the environment. (c) The results of the final structures are the optimal ones needed in the real environment, and the resulting structures are similar to existing structures in a similar real environment.

This paper proposed that the "optimization" of the self-organization process had a more profound research value. By using the characteristics of "optimization" and "emergence" to simulate the designed target structure, the environment and the possibility of stable results are created. Different from the optimization methods similar to some other programming and optimization techniques, this optimization method with a self-assembly ability has a lot of practical values and significance, which need further study and exploration.

**Author Contributions:** Conceptualization, Y.L. and S.Z.; Data curation, F.S.; Formal analysis, X.J. and Q.N.; Funding acquisition, Y.L. and S.Z.; Investigation, Y.C. and X.J.; Methodology, Y.L., X.J. and Q.N.; Software, X.J., Q.N. and F.S.; Supervision, Q.X. and S.Z.; Validation, S.Z.; Visualization, X.J., F.S. and Q.X.; Writing—original draft, Y.C.; Writing—review & editing, S.Z., Y.C. and C.L. All authors have read and agreed to the published version of the manuscript.

**Funding:** This work was supported by the National Natural Science Foundation of China under grant (No. 51605431); the Project of Department of Education of Zhejiang Province (No. Y202044011); and the Ningbo "Science and Technology Innovation 2025" major project, (No. 202002P2004).

**Institutional Review Board Statement:** Not applicable.

**Informed Consent Statement:** Not applicable.

**Data Availability Statement:** The study did not report any data.

**Acknowledgments:** We would like to express our gratitude to all those who have provided instructions, suggestions and guidance for our research.

**Conflicts of Interest:** The authors declare no conflict of interest.

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
