# Peer review of "Self-Assembly at a Macroscale Using Aerodynamics"

_applsci, doi:10.3390/app12157676_

Round 1
Reviewer 1 Report
The paper discusses self-assembly at mesoscale.
- However, it is unclear why the system is needed. The significance of mesoscale in the process has not been discussed.
- The magnetic forces employed are strong with the possibility to be broken by external forces. What forces were used and why?
- In Table 1, the unit (seconds) should be along with the conditions in the first row itself.
- The font of the equations used are not uniform.
- There are some confusing sentences like,
"Due to the local interaction of low-level particles, the general phenomenon of self-organization of macro functional structures is possible, which have been founded from the researches......"
Are the authors talking about self-assembly of low-level particles or macro functional structures.
what do the authors mean by "founded from researches.. "
Author Response
Overall Response
We appreciate the careful review. The advice has considerably contributed to improving the clarity and content of the revised manuscript. The manuscript has been checked and the corrections are marked up by using the “Track Changes” function and highlighted in yellow.
Reviewer Comment
- Comment: The paper discusses self-assembly at mesoscale. However, it is unclear why the system is needed. The significance of mesoscale in the process has not been discussed.
Response: We found that your comment is very constructive, and the significance of this self-assembly device in this manuscript has been enriched in the Second paragraph, Page2 (highlighted in yellow).
- Comment: The magnetic forces employed are strong with the possibility to be broken by external forces. What forces were used and why?
Response: Thank you for your comment. When the balls are blown by the wind, they may collide with the walls of the assembly space. And if the contact point is only on one table tennis ball at the time of colliding, the inertia force of other balls may cause the contact surface of the two magnets to slip and no longer be unconnected.
- Comment: In Table 1, the unit (seconds) should be along with the conditions in the first row itself.
Response: Thank you for your advice. In the revised version, we have removed these mistakes.
- Comment: The font of the equations used are not uniform.
Response: The font of the equations is now uniform in the revised version.
- Comment: There are some confusing sentences like:"Due to the local interaction of low-level particles, the general phenomenon of self-organization of macro functional structures is possible, which have been founded from the researches......" Are the authors talking about self-assembly of low-level particles or macro functional structures. what do the authors mean by "founded from researches…" .
Response: We are sorry for our deficient sentences and thank you for your comment. We would like to explain the meaning of these sentences you mentioned, that is: The self-organization process of macro functional structures is realized by the local interaction of low-level particles. And the feasibility of this theory has been founded on some research in other fields. And we also have made some efforts to explain well for these sentences in the revised version.
All the comments you proposed are greatly helpful for us to polish our manuscript. Thank you!

Reviewer 2 Report
This is an usual piece of research whose basis and conclusions are innapropiately developped and explained. The authors try to mimic a self-assembly procedure by using macroscopic items. In my opinion, in a self-assembly process key parameters include chemical forces between molecules, solvent, and when immovilized in surfaces, the surface nature. None of these are considered in this large scale simmulation, which only accounts for physical parameters. The obtained conclusions are vague and do not reveal any new science.
Author Response
Overall Response
We appreciate the careful review. The advice has considerably contributed to improving the clarity and content of the revised manuscript. The manuscript has been checked and the corrections are marked up by using the “Track Changes” function.
Reviewer Comment
- Comment: This is a usual piece of research whose basis and conclusions are inappropriately developed and explained. The authors try to mimic a self-assembly procedure by using macroscopic items. In my opinion, in a self-assembly process key parameters include chemical forces between molecules, solvent, and when immobilized in surfaces, the surface nature. None of these are considered in this large-scale simulation, which only accounts for physical parameters. The obtained conclusions are vague and do not reveal any new science.
Response: Thanks for your comment. In fact, the process of self-assembly not only has more research significance in the fields of chemistry, biology, etc., but also in mechanical assembly. In this article, the manufactured self-assembly machine is introduced more on mechanical assembly, and table tennis is only a case of assembled parts. Because of its spherical shape, the structures formed are compared and explained with molecules. The experimental part indeed simulates the self-assembly process of macro objects, but this article focuses more on the process of automatically generating the optimal structure from fragmented structures. As a result, the assembled objects are not limited to balls and molecules, they can also be cubes, strips, strange shapes, etc.
Since it is not limited to the self-assembly process of molecules and atoms, only basic physical parameters are considered. The results of this study may not reveal new scientific knowledge in the field of chemistry, but some relevant scientific knowledge of chemistry has been tested and verified on a macro level. The significance of the self-assembly machine introduced in this article lies more in its auxiliary role in the design process, which can automatically generate relatively stable shapes and eliminate unstable shapes.

Reviewer 3 Report
The paper of Liu et al. reports on a quite interesting experiment using table tennis balls functionalized with magnetic units. The experiment could be of some significance, however the presence of many doubtful fundamental concepts inhibits, in my opinion, the publication of this contribution in the present form. In the following, I list the more problematic issues.
1. Generally, self-assembly is the formation of supramolecular architectures from molecular building blocks. So, the nature of the interactions governing self-assembly is intermolecular. Most emphasis of the paper is given to the formation of ‘molecular’ structures.
2. A table tennis ball 40 mm in diameter and 3 gr in weight cannot be considered a mesoscopic object. It is a macroscopic object under all points of view (eq. 1-3 refer to the mechanical properties of macro objects). The term mesoscale should be avoided.
3. The term ‘down-top’ does not exist in the self-assembly literature. ‘Bottom-up’ is the term generally used.
4. I am a chemist, so I suspect that my comments suffer by some prejudices, but, in my opinion, all the reference to molecules or molecular processes are not corrected and should be avoided.
I list some problematic points:
Pg. 2, first paragraph of section 2. A table tennis ball does not resemble a molecule, that is not round in shape (a noble gas, perhaps). You can approximate CH4 as a sphere, but it is not.
Page 4, last line. This reference to the ‘octet rule’ is misleading (modern chemistry tends to avoid reference to this rule without a physical basis).
Page 5, line 7. Chemical bonding is a pure quantum effect, caused by the overlap of electronic densities. This determines stability and structure of molecules.
The nature of magnetic interactions among macroscopic bodies is a completely different story. (Magnet sites on a table tennis ball do not mimic a covalent bond). Parallel with molecules is potentially dangerous as a source of erroneous understanding.
Pg. 6, last paragraph and Figure 5. A curious inversion of logic. It is not surprising that you obtain rods, pyramids, tetrahedrons playing with magnet lego’s. Then, you call these structures CO2 (three aligned balls), H2O (three balls forming an angle) , CH4,… You are confusing the molecules and their structural properties (again coming from the chemical bonding) with their skeletal representation. These considerations are really dangerous for the comprehension of the mechanisms governing self-assembly.
5. Given the nature of this experiment, conclusions are obvious (concentration and stirring effects), and do add nothing to the understanding of self-assembly.
Author Response
Overall Response
We appreciate the careful review. The advice has considerably contributed to improving the clarity and content of the revised manuscript. The manuscript has been checked and the corrections are marked up by using the “Track Changes” function and highlighted in gray.
Reviewer Comment
- Comment: The paper of Liu et al. reports on a quite interesting experiment using table tennis balls functionalized with magnetic units. The experiment could be of some significance, however, the presence of many doubtful fundamental concepts inhibits, in my opinion, the publication of this contribution in the present form. In the following, I list the more problematic issues.
Generally, self-assembly is the formation of supramolecular architectures from molecular building blocks. So, the nature of the interactions governing self-assembly is intermolecular. Most emphasis of the paper is given to the formation of ‘molecular’ structures.
Response: Thanks for your comment. In this article, table tennis is only a case of assembled parts. And the assembled objects are not limited to balls and molecules, they can also be cubes, strips, strange shapes, etc. It was because it has a spherical shape, that the structures formed are compared and explained with molecules. In fact, the process of self-assembly not only has more research significance in the fields of chemistry, biology, etc., but also in mechanical assembly. And this manufactured self-assembly machine is introduced more on mechanical assembly. The significance of this self-assembly machine introduced in this article lies more in its auxiliary role in the design process, which can automatically generate relatively stable shapes and eliminate unstable shapes.
- Comment: A table tennis ball 40 mm in diameter and 3 gr in weight cannot be considered a mesoscopic object. It is a macroscopic object under all points of view (eq. 1-3 refer to the mechanical properties of macro objects). The term mesoscale should be avoided.
Response: We are very appreciative of your constructive advice, which made us realize that there is a misunderstanding indeed about the term ‘mesoscale’. And, the relevant description in this paper has been revised in the updated version.
- Comment: The term ‘down-top’ does not exist in the self-assembly literature. ‘Bottom-up’ is the term generally used.
Response: Thank you for your advice. In the revised version, we have made all ‘down-top’ to be ‘bottom-top’.
- Comment: I am a chemist, so I suspect that my comments suffer by some prejudices, but, in my opinion, all the reference to molecules or molecular processes are not corrected and should be avoided. I list some problematic points:
Pg. 2, first paragraph of section 2. A table tennis ball does not resemble a molecule, that is not round in shape (a noble gas, perhaps). You can approximate CH4 as a sphere, but it is not.
Page 4, last line. This reference to the ‘octet rule’ is misleading (modern chemistry tends to avoid reference to this rule without a physical basis).
Page 5, line 7. Chemical bonding is a pure quantum effect, caused by the overlap of electronic densities. This determines stability and structure of molecules.
The nature of magnetic interactions among macroscopic bodies is a completely different story. (Magnet sites on a table tennis ball do not mimic a covalent bond). Parallel with molecules is potentially dangerous as a source of erroneous understanding.
Pg. 6, last paragraph and Figure 5. A curious inversion of logic. It is not surprising that you obtain rods, pyramids, tetrahedrons playing with magnet lego’ s. Then, you call these structures CO2 (three aligned balls), H2O (three balls forming an angle) , CH4,… You are confusing the molecules and their structural properties (again coming from the chemical bonding) with their skeletal representation. These considerations are really dangerous for the comprehension of the mechanisms governing self-assembly.
Response: Thank you for your advice. The listed problematic points are explained separately as below:
Pg. 2, the first paragraph of section 2. We admit that there is a misspelling here, the term molecular is replaced by the atomic nucleus and highlighted in the revised version.
Page 4, last line. The reference to the octet rule can explain this phenomenon more simply and directly. Last, we also add your opinion from the next comment in Pg. 5 to our manuscript to reduce the possibility of misleading.
Page 5, line 7. Thank you for your opinion. We admit that there is some limitation to mimicking the chemical bonding with some magnets. We noticed that you said ‘chemical bonding ...is caused by the overlap of electronic densities.’ We agree with that. But we have to say there are similarities between electron overlap and magnet attraction, e.g., 1. they shared some communal places; 2. the closer the electron cloud is to the atomic nucleus, the stronger the electromagnetic force. The magnetic force also increases as it gets closer to the magnet. Besides, it’s hard to simulate the configuration of the extra-nuclear electron itself in detail because the peripheral electrons should even revolve around the atomic nucleus.
Pg. 6, last paragraph, and Figure 5. The paragraph is a description of Fig.5, and Fig.5 shows the structural results produced by the experiments. We all know that the volume of the nucleus is much larger than that of the electrons, so this description is not completely correct, but it is all right if only look at the composition of the nucleus. The ping pongs absolutely have differences with molecules as well as the magnets and electrons, and we admit that the description in this paragraph focuses more on the similarities rather than the differences, which is not objective enough. Finally, we have made efforts to do this part in the revised version to avoid misunderstandings.
- Comment: Given the nature of this experiment, conclusions are obvious (concentration and stirring effects), and do add nothing to the understanding of self-assembly.
Response: The results of this study may not reveal new scientific knowledge in the field of chemistry, but some relevant scientific knowledge of chemistry has been tested and verified on a macro level. And its significance of auxiliary role in the design process.
All the comments you proposed are greatly helpful for us to polish our manuscript. Thank you!

Round 2
Reviewer 3 Report
In the revised version the authors take into account most of my reliefs and changed accordingly the former version. The paper can be accepted in the present form.